# A Study on the Design of Isolator and the Mounting Method for Reducing the Pyro-Shock of a MEMS IMU

**DOI:** 10.3390/s22135037

**Published:** 2022-07-04

**Authors:** Kyungdon Ryu, ByungSu Park, Hyungsub Lee, Kyungjun Han, Sangwoo Lee

**Affiliations:** Agency for Defense Development, Daejeon 34060, Korea; byungsu_park@add.re.kr (B.P.); suby0913@add.re.kr (H.L.); kjhan@add.re.kr (K.H.); sangwoo.lee@add.re.kr (S.L.)

**Keywords:** inertial measurement unit, pyro-shock, micro electro-mechanical system

## Abstract

In this paper, we proposed two methods for reducing the pyro-shock of the MEMS Inertial Measurement Unit (IMU). First, we designed the vibration isolator for reducing the pyro-shock inside the IMU. However, it turned out that there is a limit to reducing the pyro-shock with only the vibration isolator. Therefore, we improved the pyro-shock reduction performance by changing the method of mounting on the flight vehicle. Four mounting options were tested and one of them was adopted. The results showed the best reduction performance when we designed the vibration isolator with an aluminum integrated structure. When mounting, two methods were applied. One was to insert a bracket with a different material between the mounting surface and IMU and the other was to insert a set of three washers that was stacked in a PEEK-metal-PEEK order at each part of the screw connections.

## 1. Introduction

In recent years, with the development of MEMS inertial sensor technology, research on the MEMS inertial measurement unit (IMU) is being actively carried out for application of weapon systems [1,2]. The MEMS IMU has many advantages of size, weight, power supply, power consumption, and cost compared to the mechanical and optical-type IMU. However, the MEMS IMU has a problem where a large moment of inertia cannot be obtained due to a small inertial mass. In order to overcome this, the sensor structure is designed to have a high Q factor of tens of thousands or more, and when vibrating at the resonance point, the moment of inertia can be increased regardless of the inertial mass. However, it has properties that are vulnerable to disturbances such as external vibration. The IMU is an essential component that provides the necessary information for midcourse guidance and guided control in weapon systems. Therefore, it is necessary to ensure reliability and performance of the MEMS IMU applied to a weapon system that flies with high maneuverability. On the other hand, the demand for the MEMS IMU is increasing due to the miniaturization and cost reduction of weapon systems. Therefore, various studies are being undertaken to analyze the effects of disturbances on the IMU and to improve them [3,4,5,6,7,8,9]. In order to carry out the mission, the weapon system experiences injection or separation. For this purpose, the weapon system uses a method of cutting a part of the structure or exploding the gunpowder to release the restraint. Then, a very large shock called pyro-shock is generated, and such a pyro-shock causes malfunction and performance deterioration of the weapon system components. As a result, for applying the MEMS IMU to such a weapon system, it is necessary to analyze the influence on pyro-shock and improve the shock reduction performance.

In this paper, we present two methods for reducing pyro-shock. One is an effective design method of the vibration isolator, and the other is a mounting method of the IMU on the aircraft for reducing pyro-shock. Firstly, we introduced the operating principle of the vibrating-type MEMS inertial sensor. Secondly, we compared and analyzed the various vibration isolators designed for reducing pyro-shock. Lastly, we proposed the mounting method for improving pyro-shock reduction performance and showed the results through the simulated pyro-shock test.

## 2. MEMS Inertial Measurement Unit

An IMU combines three accelerometers and gyroscopes to produce a specific force and angular rate in three-dimensions The accelerometer measures the linear acceleration of an object relative to an inertial reference frame [10,11]. The developed MEMS IMU consists of a vibratory gyroscope, a resonant accelerometer, digital board, and isolator as shown in Figure 1. The operating principles are as follows.

### 2.1. Vibratory Gyroscope

The vibratory gyroscope has a structure that is driven in the tuning fork (resonance) mode using the natural frequency of the inertial mass. When the resonance frequency of the structure is applied by the driving electrode, the structure reciprocates constantly in the drive direction (*x*-axis). When rotation (*z*-axis) is applied to an inertial mass body that is undergoing linear vibration, the Coriolis force is generated in the direction (*y*-axis) perpendicular to the vibration axis (*x*-axis) and the rotation axis (*z*-axis). The Coriolis force causes micro-vibrations in the sensing direction (*y*-axis), which have a frequency that is equal to the driving frequency and a magnitude that is proportional to the rotational input. The magnitude that is proportional to the rotation input is sensed by the sensing electrode (parallel plate electrode), and the change in the capacitance is detected to measure the applied angular velocity. As shown in Figure 2, the vibratory gyroscope has two sensors with a phase difference of 180 degrees from each other, so common error is minimized by differentiating each detected result of the angular velocity. Additionally, the structure of the vibratory gyroscope can be controlled constantly in the tuning fork mode (about 14 kHz), and therefore, it is not critically affected by the vibration disturbance equal to the frequency of tuning fork mode. However, if the structure is deformed due to a large impact, the control driver is affected, and an output can be produced abnormally.

### 2.2. Resonant Accelerometer

Resonant accelerometers are designed with a double-ended tuning fork (DTF) structure with resonators. A resonant accelerometer consists of an inertial mass and a string that structurally supports it. The accelerometer is driven by resonating the DTF strings using electrostatic force. When an acceleration is applied from the outside, the inertial mass body moves due to the inertial force. Currently, each DTF resonant string receives a tensile or a compressive force. When a tensile force is applied, the resonance frequency of the resonant string increases, and when a compressive force is applied, it decreases. The accelerometer detects the amount of change in the resonance frequency of the resonant string and converts it to the acceleration. Figure 3 is a conceptual diagram explaining the principle of the accelerometer which is called the differential resonant accelerometer. As shown in the figure, a pair of resonant accelerometers are arranged symmetrically. When the acceleration is applied to the right, the resonance frequency of DTF1 (channel A) decreases and the resonance frequency of DTF2 (channel B) increases. The applied acceleration can be detected by differentiating the amount of frequency change between them. This difference type of detection technique has the advantage of eliminating common error factors depending on the temperature of the accelerometer and increasing sensitivity [12].

The resonant accelerometer does not resonate the structure itself but uses a method of resonating the strings supporting the structure at a constant frequency and measuring the frequency change of the strings due to the movement of the structure. Due to this principle of operation, accelerometers are relatively vulnerable to external vibrations compared to gyroscopes. Figure 4 shows the frequency modal analysis results of the accelerometer structure. The environment in which the MEMS IMU operates includes the frequency range from 5 kHz to 30 kHz and more. Therefore, we analyzed up to the 12th mode which has a frequency above 40 kHz concerning a margin. It has the vibration-type drive system and high Q factor (quality factor). It is affected by vibration disturbances such as acoustic noise in the flight environment. Therefore, we designed the input frequency mode to be 20 kHz or higher, which is larger than the audible frequency range.

## 3. Design of Isolator for Reducing Pyro-Shock

The vibration-type MEMS IMU has the main frequency mode of the sensor structure distributed at 5 kHz or higher. Therefore, if a vibration disturbance with a frequency corresponding to the main mode is applied, malfunction or performance degradation may occur. The pyro-shock generated by the explosion of gunpowder has a very large transition and a very short time delay. Such pyro-shock has high g value of tens of thousands and a high frequency band of 30 kHz or more. Figure 5 shows when the pyro-shock occurred in an actual operating environment and the change in acceleration output due to the pyro-shock. In this operating environment, the 1st pyro-shock was generated by a gunpowder explosion between 0.05 to 0.15 s, and the 2nd pyro-shock was generated by the ignition motor output between 0.3 and 0.4 s. If the developed accelerometer operates ideally, the accelerometer frequency output of both channels should be a perfectly symmetrical shape as shown in Figure 6. However, the pyro-shock generates a disturbance that is equal to the main frequency mode of the IMU in general. Therefore, it is necessary to make efforts to reduce the pyro-shock in order to prevent malfunction and improve the reliability. The vibration isolator was designed inside the IMU to block a high-frequency disturbance such as pyro-shock. The vibration isolator was mounted between the upper and lower housings and was designed by considering the sensor bandwidth and operating environment.

### 3.1. Improvement Process of Isolator

The vibration isolator for reducing pyro-shock was improved and designed from the four-point fixed structure to an aluminum integrated structure via a stainless steel (SS) integrated structure and a brass integrated structure. In this section, we perform the pyro-shock test according to the design change process of the vibration isolator and explain the performance analysis and improvement process.

#### 3.1.1. Four-Point Fixed Isolator

Figure 7a shows the initial structure of the vibration isolator. The six sensor blocks (electronic board including a sensor) can be individually fastened to four points inside the vibration isolator structure, and the vibration isolator’s natural frequency is designed at 1 kHz.

Figure 8 shows the accelerometer output results from the test in the actual environment. It applied the IMU with the four-point fixed vibration isolator. It is the A and B channel frequency output results of the accelerometer in the *x*-axis, and the results are shown by inverting the B channel upside down for easier comparison of the output values of the two channels. For the ideal case, the frequency outputs of both channels should be the same. However, the frequency output of channel A could not follow the output of channel B and operated abnormally due to the 1st pyro-shock. We confirmed that the applied vibration isolator could not sufficiently reduce the pyro-shock.

#### 3.1.2. SS Integrated Isolator

The four-point fixed vibration isolator has a high natural frequency and a low reduction effect in the high-frequency range due to its small mass. As a result, the pyro-shock is transmitted to the sensor board without sufficient reduction. Therefore, in order to reduce the external vibration disturbance and shock energy, we designed an integrated vibration isolator that filled the central region of the sensor block with the same material as shown in Figure 7b.

The integrated vibration isolator was designed to have a lower natural frequency than the four-point fixed vibration isolator. Figure 9 shows the results of comparing the frequency response characteristics of the four proposed isolators that are modeled with a simple mass-spring-damper system as shown in Figure 10. From this result, it could be inferred that the integrated structure has a greater reduction effect in the high-frequency range than the four-point fixed structure in the shock range of 5 kHz or more.
(1)$$m+\dot{\delta }+k_{tot}\delta =-\ddot{u}$$

Figure 11 shows the results of the A and B channels of the accelerometer after performing a flight test using the IMU with the stainless-steel integrated isolator. It operated normally in the 1st pyro-shock section, which showed a malfunction before. However, there still existed a large error in the 2nd pyro-shock section. Therefore, additional improvements of the isolator were needed to ensure the reliability of the IMU.

#### 3.1.3. Brass Integrated Isolator

We simply changed the material of the isolator to brass, which has a relatively higher density than stainless steel, while maintaining the shape in order to improve the shock reduction performance of the vibration isolator. As a result of a flight test using a brass integrated isolator, it was confirmed that the shock reduction performance was higher than that of the stainless-steel isolator as shown in Figure 12.

#### 3.1.4. Aluminum Integrated Isolator

The effect of the brass integrated isolator was verified by the pyro-shock test. However, there are some disadvantages to using brass. First, the weight of the IMU itself increases due to the high density. Second, the high unit price of brass itself can increase the unit cost of production and reduce productivity in the future. Therefore, aluminum was adopted as the material of the vibration isolator that has a relatively low density and a low unit price. Before the pyro-shock test, we performed a frequency response test using a high-frequency vibration testing device as shown in Figure 13a. While applying the acceleration with a magnitude of 1 g and a frequency range of 10 to 50 kHz, the reduction amount before and after passing through the vibration isolator was compared in the frequency domain. From the result of the oscilloscope in Figure 13b, it could be confirmed that aluminum has a greater reduction effect than brass in the pyro-shock frequency range of about 25 to 35 kHz. Finally, the aluminum integrated isolator as shown in Figure 7d was applied to the IMU.

## 4. Mounting Method for Reducing Pyro-Shock

After applying the IMU with the aluminum integrated isolator, it seemed that the shock reduction performance was verified through numerous flight tests. However, in the flight test in which a larger pyro-shock was accidentally applied than before, there was a problem that the *x*-axis accelerometer malfunctioned as shown in Figure 14. If the accelerometer is operating normally, the output frequencies of channels A and B must match each other, but it malfunctioned in the 1st pyro-shock. Based on this phenomenon, we tried to find a method that could further reduce the pyro-shock.

### 4.1. Description of the Four Mounting Methods

As a method for reducing the pyro-shock, we devised a simulated pyro-shock test for four mounting methods as shown in Table 1. The reduction performance was compared and analyzed for each method through the pyro-shock simulator. Case #1 is the method of directly fastening to the mounting surface using bolts, which is the original mounting method. In this case, external noise and shocks are transmitted to the sensors directly through the mounting surface. Case #2 is the method of inserting a PEEK washer between the fastening part and the bolt to float between the mounting surface and the IMU so that the external disturbances are transmitted only through the fastening part. For Case #3, a bracket of a different material was inserted between the mounting surface and the IMU, and PEEK washers were inserted at all fasteners. In Case #4, a metal washer was added between the two PEEK washers at the fastening part. This was based on a study that reduced the pyro-shock in the same way [13,14].

### 4.2. Multi-Layered (PEEK-Steel-PEEK) Isolator

An isolator made of rubber or a wire rope, as shown in Figure 15, has a typical transmission rate characteristic [15]. Therefore, the reduction effect of the isolator appears in the frequency region that is 1.4 times or more of the natural frequency. A better reduction effect appears in the higher frequency region. However, such isolators have less of a reducing effect in the low frequency region, which can cause a measurement error of the IMU. The multi-layered (PEEK-steel-PEEK) isolator used in the proposed mounting method (Case #4) could reduce not only the shock with high-frequency components but also the shock with low-frequency components. The principle of such a mounting method is that when washers made of different materials are sandwiched alternately, it results in the effect of reflecting the shock energy transmission due to the difference in stiffness between the two materials. Furthermore, due to the washer stacking, the surface contact characteristic between the mounting surface and the inertial measuring instrument is changed to the point contact characteristic, and the shock reduction effect due to the reduction in the impact transmission area is also exhibited. Since the PEEK washer is a polymer material and has a stiffness of about 5% of that of a general steel material, it was judged that the shock reduction effect due to the difference in stiffness is good. In addition, it has good thermal characteristics and is guaranteed to be stable even at the required tightening torque value, so it is suitable for systems that require reliability in extreme environments such as guidance missiles.

### 4.3. Reduction Effect Analysis through Pyro-Shock Test

#### 4.3.1. Environment

We used the pyro-shock simulator in the Korea Aerospace Research Institute (KARI) for the comparison and analysis of the pyro-shock reduction performance of the four mounting methods. Figure 16a shows the pyro-shock simulator and Figure 16b shows the IMU mounted on the pyro-shock simulator. Table 2 shows the specification of the simulator. A pyro-shock can be simulated by hitting a heavy iron ball on the top plate using an air compressor.

#### 4.3.2. Results

Figure 17 shows the SRS results for the pyro-shock test. As shown, the shock generated using the simulator is like the actual pyro-shock in magnitude and frequency, which has a magnitude of 1000 g or more and a frequency at 1 kHz or higher. The amplification phenomenon was observed around 400~600 Hz because the natural frequency of the vibration isolator exists within the range. In all the cases, it could be confirmed that the shock was reduced by the vibration isolator in the frequency band of 800 Hz or higher. Since there was no difference in the reduction performance between Case #1 and Case #2, it was found that using the PEEK washer alone was not effective. In Case #3, the reduction performance was relatively low in the 3 to 7 kHz band. It was probably due to the influence of the inserted bracket. Case #4 had the best reduction performance among the all-mounting methods especially at 5 kHz or higher, which is estimated to be the pyro-shock frequency band. From this result, inserting two different types of washers in order of PEEK-metal-PEEK was found to be the most effective method. Therefore, we finally adopted and applied the Case #4 mounting method.

## 5. Conclusions

In this paper, we proposed two methods for reducing the pyro-shock, which was applied to IMU. One was the design method of the vibration isolator, and the other was the mounting method of the IMU. The shock reduction performance was compared and analyzed by the high-maneuvering flight test and the simulated pyro-shock test. The adopted vibration isolator had an integrated aluminum structure which had a low resonance frequency. It showed effective shock reduction performance in the actual pyro-shock test, which had a frequency band from 25 to 35 kHz. While performing several flight tests, however, a problem occurred where the accelerometer malfunctioned due to a large pyro-shock than before under certain flight-environmental conditions. Therefore, we additionally supplemented the original mounting method of the IMU for reducing the pyro-shock and ensuring the reliability. The simulated pyro-shock tests were conducted for the four mounting methods. As a result, we found that the best method was applying the mounting bracket between the mounting surface and the IMU and then inserting the two different types of washers to all the fastening parts in the order of PEEK-metal-PEEK. Finally, we ensured the reliability of the improved IMU by conducting several flight tests in a high-maneuvering environment.

## Figures and Tables

**Figure 1 sensors-22-05037-f001:**
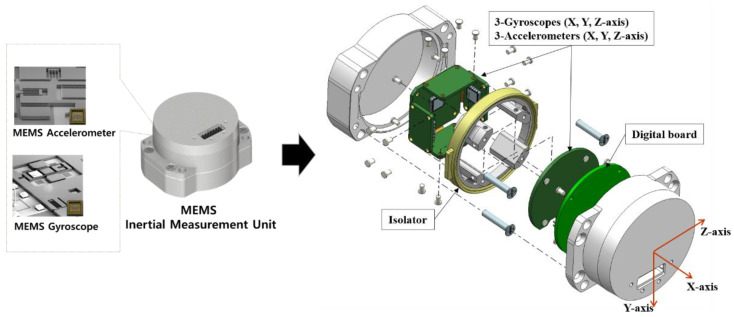
The detailed description of MEMS IMU which was used for pyro-shock test.

**Figure 2 sensors-22-05037-f002:**
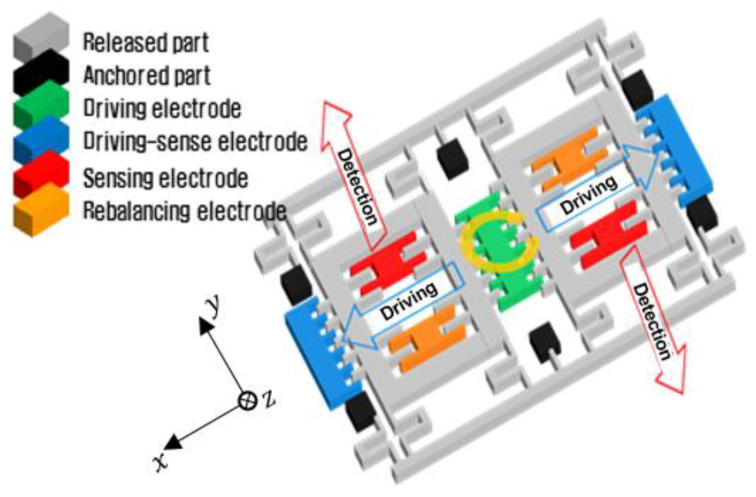
Structure of tuning fork vibratory gyroscope.

**Figure 3 sensors-22-05037-f003:**
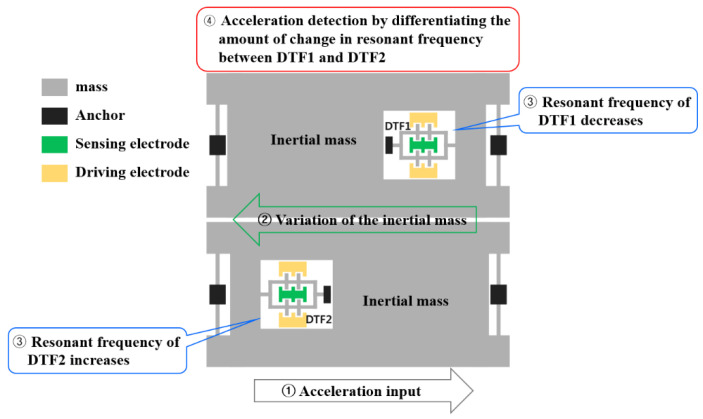
Conceptual diagram of the principle of detecting acceleration.

**Figure 4 sensors-22-05037-f004:**
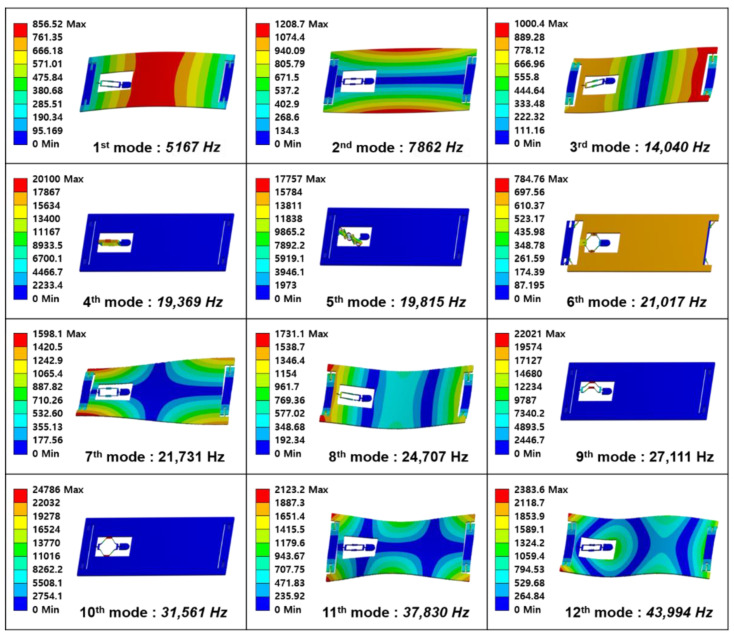
Modal analysis of the structure of the MEMS IMU on the frequency range of about 5~40 kHz that corresponds to the operating environment.

**Figure 5 sensors-22-05037-f005:**
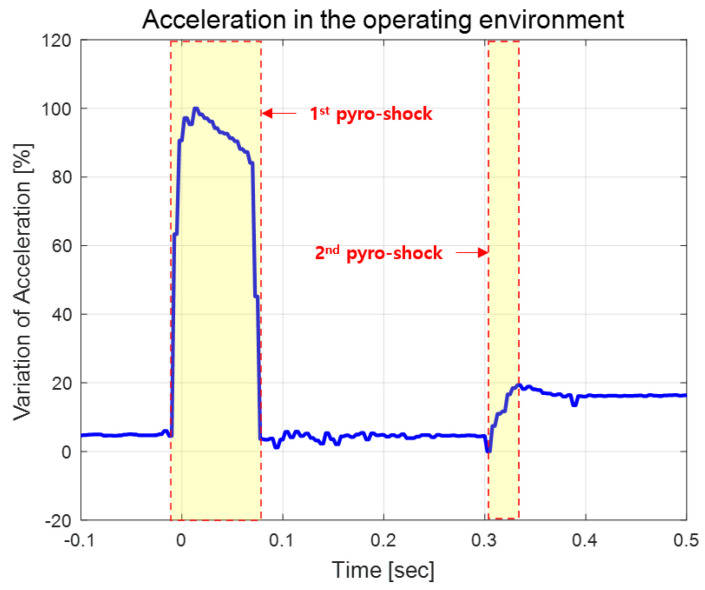
Acceleration profile in actual operating environment.

**Figure 6 sensors-22-05037-f006:**
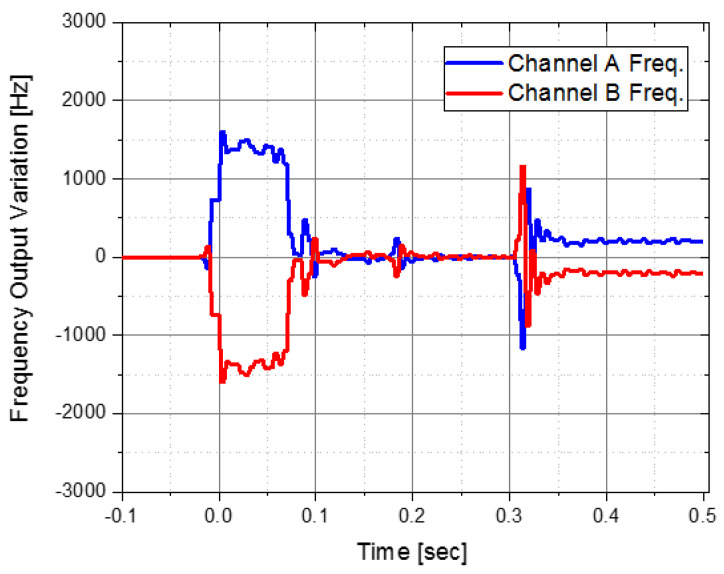
Ideal frequency output for two channels.

**Figure 7 sensors-22-05037-f007:**
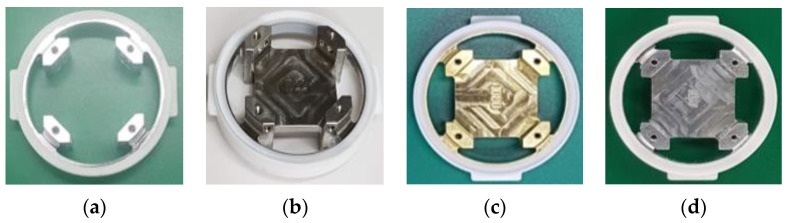
Improvement process of designed vibration isolator: (**a**) 4-point fixed isolator; (**b**) stainless steel integrated isolator; (**c**) brass integrated isolator; (**d**) aluminum integrated isolator.

**Figure 8 sensors-22-05037-f008:**
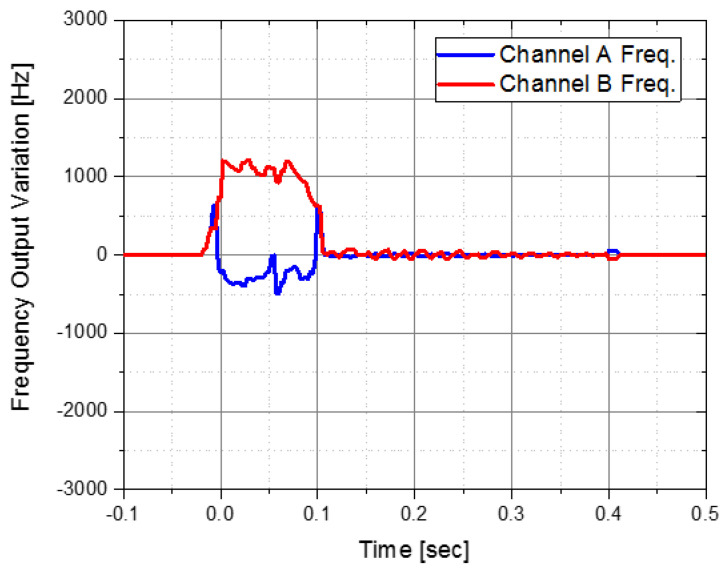
Accelerometer frequency output from IMU with 4-point fixed isolator.

**Figure 9 sensors-22-05037-f009:**
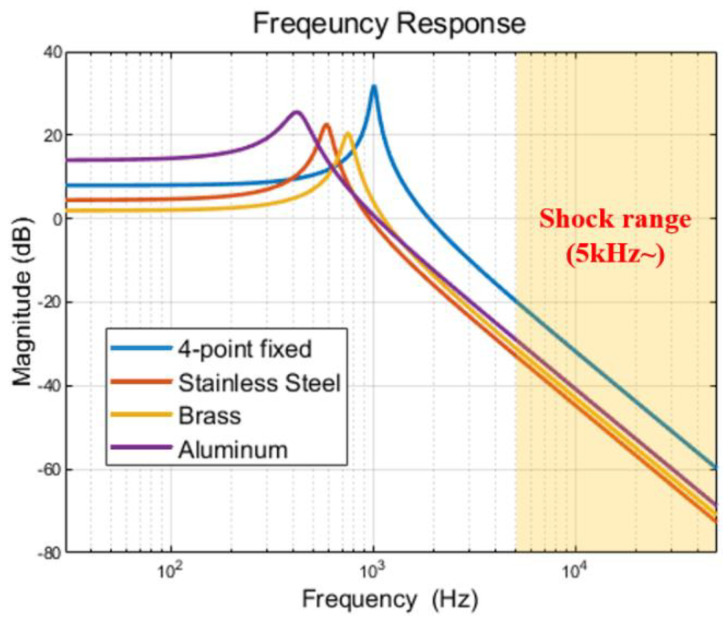
Frequency response between 4-point fixed isolator and stainless-steel integrated isolator.

**Figure 10 sensors-22-05037-f010:**
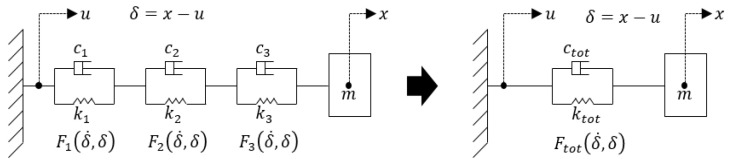
Linear mass-spring-damper system for simulating the 4 isolators (4-point fixed, stainless steel, brass, and aluminum integrated).

**Figure 11 sensors-22-05037-f011:**
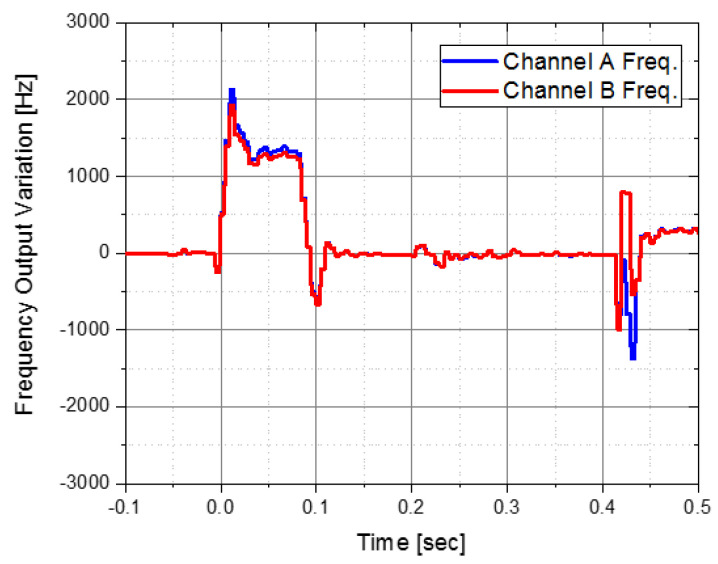
Accelerometer frequency output from IMU with stainless steel integrated isolator.

**Figure 12 sensors-22-05037-f012:**
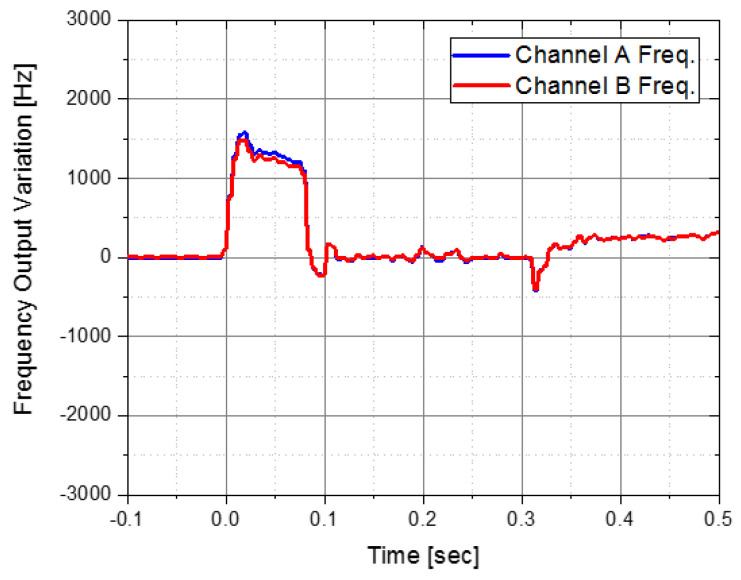
Accelerometer frequency output from IMU with brass integrated isolator.

**Figure 13 sensors-22-05037-f013:**
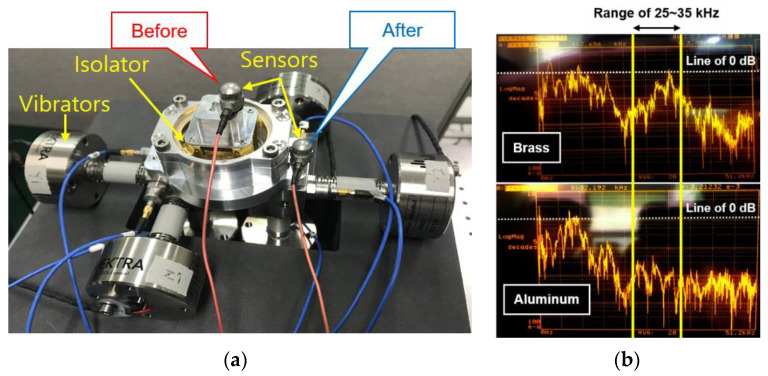
Test using high-frequency vibration equipment: (**a**) equipment; (**b**) oscilloscope result.

**Figure 14 sensors-22-05037-f014:**
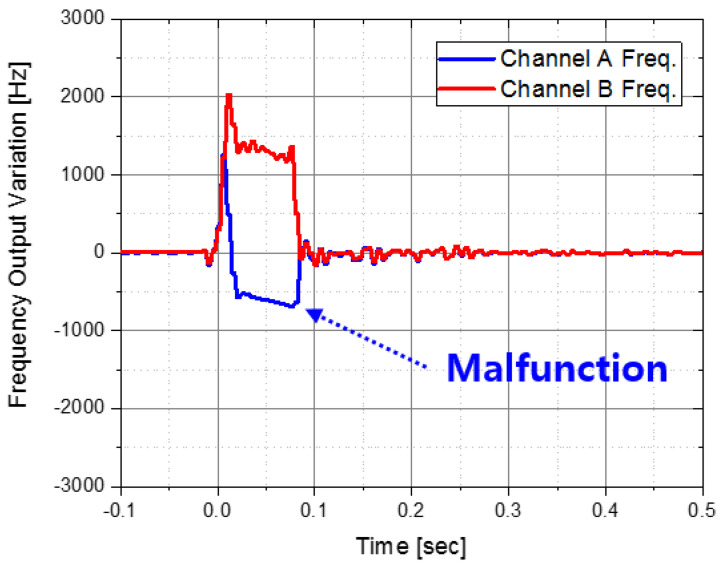
Malfunctioning case of accelerometer.

**Figure 15 sensors-22-05037-f015:**
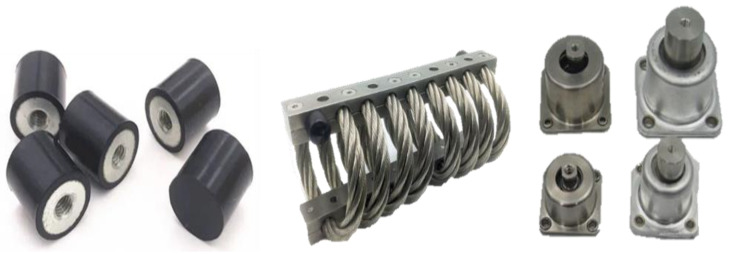
Typical shock isolators made of rubber, spring, etc.

**Figure 16 sensors-22-05037-f016:**
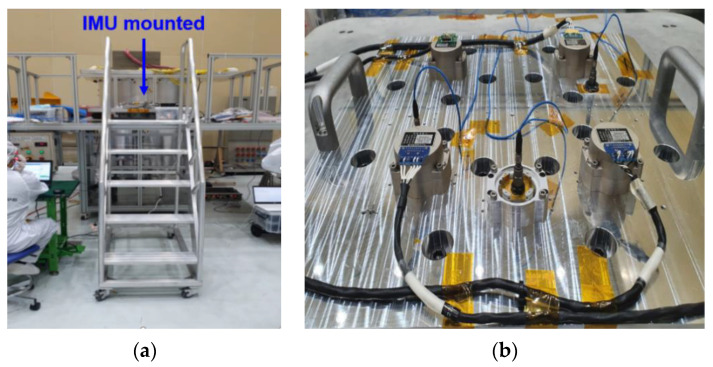
Simulated pyro-shock testing environment: (**a**) pyro-shock simulator; (**b**) top view of pyro-shock simulator.

**Figure 17 sensors-22-05037-f017:**
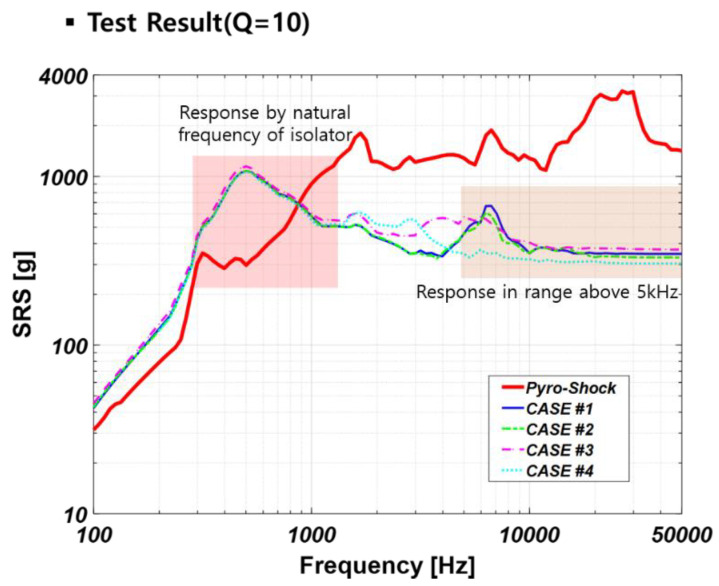
SRS result of pyro-shock simulation.

**Table 1 sensors-22-05037-t001:** Four methods applied for reducing pyro-shock.

Case	Cross-Section	Description
#1	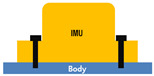	Original mounting method Direct fastening to the mounting surface
#2	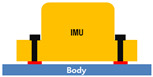	Inserting PEEK washers between the mounting surface and the IMU
#3	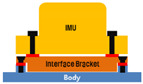	Inserting PEEK washers to all the fastening parts Inserting a bracket of a different material between the mounting surface and the IMU
#4	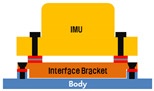	Inserting PEEK washers to all the fastening part Inserting a bracket of a different material between the mounting surface and the IMU Add a metal washer between the two PEEK washers at each fastening part (in order of PEEK-metal-PEEK)

**Table 2 sensors-22-05037-t002:** Specification of pyro-shock simulator.

Specification of Pyro-Shock Simulator	
Maximum load	150 kg
Shock area	1 m×1 m
Natural frequency of top plate	1000~1500 Hz
Maximum shock	2000 g (Q=10)

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
