# Peer review of "A Study on the Design of Isolator and the Mounting Method for Reducing the Pyro-Shock of a MEMS IMU"

_sensors, 2022, doi:10.3390/s22135037_

Round 1

Reviewer 1 Report

The recommendations/comments are embedded in the attached manuscript.  

Reviewer 2 Report

Authors’ research proposed two methods for reducing pyro-shock of the MEMS IMU. First, authors designed the vibration isolator for reducing pyro-shock inside the IMU. Then, authors improved the reduction performance of pyro-shock by changing the method of mounting on the flight vehicle. As a result, it showed the best reduction performance when authors designed the vibration isolator with aluminum integrated structure, which is very innovative and interesting. However, some comments as follows need to be explained or revised.

1. In order to demonstrate the superiority and innovation of authors' method, authors need to discuss and compare the isolator and the mounting method authors proposed with the traditional methods, and determine whether the effect is better.

2. In Chapter 2, it is introduced the operating principle of the vibrating type MEMS inertial sensor. The modal analysis was also carried out. As you know, MEMS inertial measurement units main frequency mode depends on its structural parameters. Authors analysis lacks key analysis parameters here.

3. The background words in Figure 4 are too small, it is recommended to redraw the background.

4. In Chapter 3, authors compared and analyzed the various vibration isolators (4-points fixed isolator, Stainless-integrated isolator, Brass-integrated isolator and Aluminum integrated isolator) designed for reducing pyro-shock. However, there is a lack of theoretical analysis or mathematical models based on the those isolators.

5. Figure 12(b) need to be clearly drawn.

6. As analyzed by authors, the tuning fork mode of the vibratory gyroscope is about 14kHz. The main frequency modes of the accelerometer structure exist in the high frequency range of 5kHz and above. Authors need to pay more attention to vibration isolation effects in those frequency range.

7. The first occurrence of an abbreviation needs to be explained, such as SUS in line 132, and SRS in line 240.

Reviewer 3 Report

This work presents design of the isolator and mounting method to reduce the pyro-shock of MIMU. It is a certain valuable for some special technical engineer in this field. However, lots of issues should be clarified to satisfy the demand of this Journal. The detailed comments are given as follows.

1, Mounting method is important to improve anti-shock ability. Please add the corresponding theory to support the experimental test. 

2, It's a little meaningful to list lots of analysis on different vibration isolator if pyro-shock reduction with only the isolator is very limited. Advising to add more mounting method analysis.

3, Improve readability of refer. in the introduction to ensure the advance and significant of this research.

4, lots of pictures are unclear.
